# The Effect of GD1a Ganglioside-Expressing Bacterial Strains on Murine Norovirus Infectivity

**DOI:** 10.3390/molecules25184084

**Published:** 2020-09-07

**Authors:** Yifan Zhu, Hiroki Kawai, Satoshi Hashiba, Mohan Amarasiri, Masaaki Kitajima, Satoshi Okabe, Daisuke Sano

**Affiliations:** 1Department of Frontier Sciences for Advanced Environment, Graduate School of Environmental Studies, Tohoku University, Aoba 6-6-06, Aramaki, Aoba-ku, Sendai, Miyagi 980-8579, Japan; zhu.yifan.t1@dc.tohoku.ac.jp; 2Division of Environmental Engineering, Faculty of Engineering, Hokkaido University, North 13 West 8, Kita-ku, Sapporo, Hokkaido 060-8628, Japan; cross.road836769@gmail.com (H.K.); s.hashiba426@gmail.com (S.H.); mkitajima@eng.hokudai.ac.jp (M.K.); sokabe@eng.hokudai.ac.jp (S.O.); 3Department of Civil and Environmental Engineering, Graduate School of Engineering, Tohoku University, Aoba 6-6-06, Aramaki, Aoba-ku, Sendai, Miyagi 980-8579, Japan; mohan@kitasato-u.ac.jp; 4Department of Health Science, School of Allied Health Sciences, Kitasato University, 1-15-1 Kitasato, Sagamihara-Minami, Kanagawa 252-0373, Japan

**Keywords:** GD1a ganglioside, murine norovirus, specific virus–cell interaction, virus environmental fate, enzyme-linked immunosorbent assay, receptor glycan

## Abstract

In this study, we investigated the impact of GD1a-expressing bacterial strains on the infectivity of murine norovirus (MNV). Eligible bacterial strains were screened from a sewage sample using flow cytometry, and their genetic sequences of 16S rRNA were determined. The enzyme-linked immunosorbent assay (ELISA) was employed to analyze the binding between bacteria and MNV particles, and the plaque assay was used to assess the effects of GD1a-positive and negative strains on MNV infectivity. The result from ELISA shows that MNV particles are able to bind to both GD1a-positive and negative bacterial strains, but the binding to the GD1a-positive strain is more significant. The infectivity assay result further shows that the MNV infectious titer declined with an increasing concentration of GD1a-positive bacteria. The addition of anti-GD1a antibody in the infectivity assay led to the recovery of the MNV infectious titer, further confirming that the binding between MNV particles and bacterial GD1a ganglioside compromises MNV infectivity. Our findings highlight the role indigenous bacteria may play in the lifecycle of waterborne enteric viruses as well as the potential of exploiting them for virus transmission intervention and water safety improvement.

## 1. Introduction

Unsafe water poses a serious threat to public health worldwide. A global health risk assessment program estimated that unsafe water sources led to 1.2 million deaths and the loss of 71.7 million disability-adjusted life years in 2015 [1]. Among the various risk factors that compromise water safety, human enteric viruses are a significant concern and pose a serious burden to global health care systems [2,3,4]. For instance, it was estimated that noroviruses, a group of non-enveloped icosahedral viruses from the *Caliciviridae* family and a leading cause of acute gastroenteritis [5,6,7], cause 70,000 to 200,000 deaths per year on a global scale [8,9]. Common clinical aftermaths resulting from human enteric virus infection range from relatively mild symptoms, such as vomiting and diarrhea, to some serious, even fatal, diseases, including encephalitis, paralysis, and aseptic meningitis, depending on the immune status and age group of the infected individual [10,11].

The primary source of human enteric viruses in aquatic environments is discharge from infected individuals [12,13]. For human norovirus, as an example, a shedding rate as high as 1.65 × 10^12^ genome copies per gram of feces from infected individuals has been reported [14]. Human norovirus infection is mostly self-limiting, and the clinical symptoms generally resolve within a few days, but the prolonged viral RNA discharge can last for up to 2 months under certain conditions [15,16,17]. This intensive and persistent shedding, coupled with the fact that enteric virus particles can be highly resistant to environmental factors after being released into aquatic environments [10,13,18,19,20], results in the ubiquitous prevalence of human enteric viruses in various bodies of water, including surface water, groundwater, seawater, and sewage [3].

Pang et al. [21] traced the prevalence of seven human enteric viruses in river water. In their study, rotavirus was most frequently detected with an 87% detection rate, and its concentration ranged from 2.3 to 4.5 log_10_ genome copies per liter of river water. Other viruses such as norovirus and sapovirus, on the other hand, showed a clear seasonal pattern, with the peak appearing in winter months. In the case of sewage, due to its close link to fecal content, a high level of enteric virus concentration has often been recorded. Eftim et al. [22] reviewed some previous wastewater norovirus quantification studies and reported a mean concentration of 4.9 log_10_ genome copies of norovirus GII per liter of raw sewage, followed by a slightly lower, but still considerable, occurrence of norovirus GI of 4.4 log_10_ genome copies per liter of raw sewage. In addition to this high prevalence, many human enteric viruses also have a low infectious dose, so a small amount of virus particles, often in the range of tens to hundreds, is enough to cause infection [12,23]. With all these factors combined, once the drinking water supply or water reclamation system has malfunctioned, the result can be acute gastroenteritis outbreaks caused by the contamination of drinking water by sewage leakage [24,25], and food poisoning caused by consuming crops irrigated with inadequately treated wastewater [26,27].

To achieve better control of the outbreaks of waterborne human enteric virus infection, a crucial step is to build a solid understanding of the fate of these viruses in aquatic environments and the role such fate plays in the transmission cycle. In-depth research can facilitate the process of conceiving novel strategies for risk management and transmission pathway intervention [28].

The interaction between enteric virus particles and substances in aquatic environments can be divided into two groups: non-specific and specific. Non-specific interaction chiefly consists of inter-molecular forces such as van der Waal’s forces, long-range electrostatic interaction, and the hydrophobic effect [29,30,31], whereas specific interaction generally refers to receptor–ligand recognition and binding effects, which tend to have a strain-dependent specificity [32,33,34]. In recent years, many studies have focused on the relatively understudied yet rather important specific interaction, which can have an impact on many aspects in the enteric virus transmission pathway, such as environmental survivability, the efficiency of removal and inactivation during treatment, infectivity within hosts, etc. [35]. Therefore, a deeper look into the virus–bacterium interaction profile is essential for a more comprehensive interpretation of its nature and the influencing factors, which could potentially inspire improvements in sanitation strategies and water safety planning.

The virus infection process is initiated by the attachment of the virus particle to the target host cells [32,34,36,37]. Because glycans form the foundation of a variety of biological functions in living cells and can be found in abundance on cell surfaces, viruses, not surprisingly, have gradually evolved various strategies to take advantage of these substances by using them as specific receptors to assist in their attachment to cells. Glycans are recognized by more than half of all mammalian virus families as receptors, and thus, substantial efforts have been expended to investigate the nature of glycan–virus interaction [38,39]. The most studied glycan receptor to date is histo-blood group antigens (HBGAs), a series of complex carbohydrates that can be found on mucosal epithelial cells or in secretions such as the blood and saliva as free antigens, which can be recognized and utilized by human norovirus strains for the infection process in a genotype-specific manner [39,40,41,42,43,44,45,46]. Other than HBGAs, specific virus-binding proteins [47,48], heparan sulfate proteoglycan [49], and sialic acid (SA) linked to glycoproteins and gangliosides [11] are also known to support the infection processes of a wide spectrum of viruses [41,45,50]. Murine norovirus 1 (MNV-1), for instance, recognizes terminal SA moieties in GD1a ganglioside to infect murine macrophages [51].

The impact of specific virus–bacterium binding on the environmental fate of waterborne viruses, although already addressed in some previous studies, is still unclear. Li et al. [52] have suggested that binding shields virus particles from heat treatment, thereby enhancing their environmental persistence and resistance to treatment processes. Under certain circumstances, however, binding can also be beneficial for restraining the spread of viruses. Several studies have acknowledged that virus removal during the wastewater treatment process can be improved by the interaction of virus particles with the biomass [53,54,55,56], because the bound virus particles are easier to remove from the effluent by sedimentation or membrane filtration. Nevertheless, to what extent specific binding contributes to this characteristic is not fully understood at this point and needs further investigation.

Our current study aims at expanding the existing knowledge base regarding the specific virus–bacterium interactions by focusing on the effects that specific receptor-expressing bacterial strains in municipal water supply systems can have on virus–bacterium binding and the viruses’ infectivity profiles. GD1a ganglioside was selected as the subject receptor for its role in MNV binding. First, we designed and performed a serial screening approach with the intention of identifying and isolating indigenous GD1a-positive bacterial strains from a sewage sample based on their interaction with anti-GD1a antibody. The respective species of the bacterial strains were then identified through 16S rRNA gene sequencing. In the next phase of the experiment, one such strain was featured in the ELISA and plaque assay using MNV as the surrogate for human norovirus, the objective being to evaluate whether GD1a ganglioside expression makes a difference in the MNV–bacterium binding profile and MNV infectivity for RAW 264.7 cells.

## 2. Results

### 2.1. The Western Blotting Result for GD1a Expression Confirmation

The expression of GD1a ganglioside in tightly bound extracellular polymeric substances (TB-EPS) of the bacterial strains screened from the sewage sample was examined by Western blotting (Figure 1). Of the four strains tested, strain B4 showed two bands at around 20 and 33 kDa, and therefore was chosen to be the GD1a-positive strain used in the subsequent experiments. The following 16S RNA sequencing revealed that strain B4 belongs to the species *Stenotrophomonas maltophilia*. By contrast, strain F5, belonging to the species *Chryseobacterium massiliae*, did not show any sign of obvious banding or smearing and hence was used as the GD1a-negative strain for control.

### 2.2. The Effect of GD1a-Expressing Bacterial Strains on the MNV Binding Profile

The extent of adsorption between MNV and bacteria was assessed by ELISA with the anti-GD1a antibody (Figure 2). The extent of virus–bacterium binding was measured according to relative adsorption intensity. A relative adsorption intensity of 1.0 means the addition of the MNV suspension does not affect the amount of anti-MNV antibody bound to the plate well surface, and, therefore, the binding between MNV particles and the immobilized bacterial cells cannot be confirmed. By contrast, a value larger than 1.0 means the anti-MNV antibody is more effectively bound to the well surface thanks to the MNV suspension being added, thereby confirming that the MNV particles do bind to the immobilized bacterial cells. The relative adsorption intensity values of all four MNV–bacterium combinations were larger than 1.0, suggesting that both bacterial strains aided the MNV binding capability to some extent, regardless of the GD1a-expressing capability. However, *S. maltophilia* B4 demonstrates a significantly (*p* < 0.05) higher relative adsorption intensity (26.7% for MNV S7 and 14.8% for MNV-1) than *C. massiliae* F5 for both MNV strains.

The results of the plaque assay (Figure 3) are in line with those of the ELISA. Both MNV strains showed significantly (*p* < 0.05) lower infectious titers in the presence of *S. maltophilia* B4 in comparison with *C. massiliae* F5. To be precise, the titer was 58.4% lower in the case of MNV-1 and 58.1% lower for MNV S7. The result indicates that *S. maltophilia* B4 has an edge over *C. massiliae* F5 in reducing unbound and free-moving virus particles. Additionally, with either strain, the recorded virus infectious titer was somewhat lower than that of the control group, corroborating the universal binding effect unrelated to GD1a-positivity shown in the ELISA result.

### 2.3. The Effect on MNV Infectivity

When centrifugation of the mixture of bacterial strains and MNV stock is not performed prior to the inoculation step, the virus particles bound to bacterial cells are not physically removed and are still theoretically capable of infecting RAW cells if the infectivity has not been compromised by the binding to bacterial cells. Thus, the virus infectious titer should serve as an indicator of virus infectivity (Figure 4). For the combination of MNV-1 and *S. maltophilia* B4, infectious titers were lower than those for the control group when the bacterial concentration was 2.5 × 10^5^ cells/mL and below, indicating that the inhibitory effect of *S. maltophilia* B4 on MNV-1 infectivity starts from a low concentration. As the cell concentration increased, the effect became more pronounced, and when the cell concentration reached 4.0 × 10^6^ cells/mL and beyond, no plaque could be counted, meaning that a high concentration of *S. maltophilia* B4 significantly compromised MNV-1 infectivity. *C. massiliae* F5, on the other hand, showed a minor inhibitory effect on MNV-1 infectivity, as plaques could be counted under all the tested bacterial concentration levels. Even under the highest bacterial concentration employed in this study (2.54 × 10^8^ cells/mL), the MNV-1 infectious titer dropped by only 34% compared to that in the control group, in which the bacterial suspension was not added.

As for MNV S7, a decrease in the infectious titer was also observed when *S. maltophilia* B4 was added to the cell culture at low concentrations. When the added concentration of *S. maltophilia* B4 reached about 1.0 × 10^3^ cells/mL, the MNV infectious titer saw a drop of about 44% from that of the control group, and no plaque formed under bacterial concentrations of 1.6 × 10^7^ cells/mL or higher. By comparison, the presence of *C. massiliae* F5 did not seriously affect MNV S7 infectivity, reflected by the fact that the infectious titer stayed high across the bacterial concentration range, with the exception of 2.55 × 10^8^ cells/mL, the highest bacterial concentration tested, under which no plaque formed. At 6.37 × 10^7^ cells/mL, the infectious titer dropped by only 30% compared to that of the control group.

Finally, the plaque assay result with the addition of anti-GD1a antibody is shown in Figure 5. The two antibody concentrations tested in this study yielded similar results. As the bacterial concentration rose from zero to about 1.0 × 10^6^ cells/mL, the virus infectious titer stayed relatively stable, but when the bacterial concentration continued to increase and reached the level of 4.0 × 10^6^ cells/mL, the plaque count significantly dropped for both MNV strains (98% for 0.02 ug/mL antibody addition, and 78% for 2.5 ug/mL antibody addition, *p* < 0.05), and further increasing the bacterial concentration almost abolished the formation of plaques.

## 3. Discussion

In this study, by performing ELISA and infectivity assays, we found that the GD1a-positive bacterial strain shows a better MNV particle binding ability compared with the negative strain, and the GD1a-positive strain compromises MNV infectivity more than does the negative strain when not physically separated from the mixture by centrifugation prior to inoculation. The function of GD1a ganglioside was also corroborated by the subsequent antibody experiment, because the addition of the anti-GD1a antibody effectively restored MNV infectivity under low to medium bacterial concentration levels, but as the bacterial concentration continued to increase, the infection was still suppressed.

Gangliosides are ubiquitously present in vertebrate cells and are involved in a handful of biological functions, notably in the nervous system [51,57,58], but little is known regarding other aspects of these substances. By employing the serial screening approach, the present study confirms that bacterial strains that can express GD1a ganglioside, which has previously been found to contribute to MNV–cell binding, can be found in aquatic environments. It is worth mentioning that GD1a-positive bacterial strains are not the only ones found in aquatic environments that can produce specific virus receptors. Amarasiri et al. [59] successfully identified non-human HBGA-bearing bacterial strains in a mixed liquor suspended solids (MLSS) sample taken from a pilot-scale membrane bioreactor. They also found that the virus removal performance of membrane filtration was enhanced in the presence of these bacteria via more efficient membrane separation, although the efficacy is likely to be also dependent on other factors such as the virus genotype and the location of the HBGAs. The reason for the occurrence of receptor-bearing bacteria in the aquatic environment remains unknown, but it has been postulated that HBGA-bearing bacteria can take advantage of this trait to avoid attacks from phagocytic cells and inhabit the intestinal track via systemic immune ignorance [60]. From the perspective of virus spreading, the environmental presence of specific receptor-expressing organisms, despite showing potential for improving virus removal efficiency [59], may also affect human health in a negative manner. One example is the bioaccumulation of human norovirus in oyster tissue due to its ability to produce an A-like carbohydrate structure that resembles human blood group A antigen [61], which boosts the risks from oyster consumption. Future studies focusing on this topic can shed new light on the environmental fate of waterborne viruses and illuminate novel approaches to interfering with human enteric virus transmission via the aquatic environment, thereby eventually better protecting public health.

From the results of the ELISA and plaque assays with centrifugation (Figure 2 and Figure 3), we conclude that *S. maltophilia* B4 does exhibit enhanced MNV binding compared to *C. massiliae* F5, reflected by both the higher relative adsorption value resulting from MNV particles bound to immobilized bacteria, and the lower virus titer due to the physical removal of MNV particles from the virus–bacterium mixture through centrifugation. It is worth noting that MNV S7 showed a higher binding level in ELISA, even though the primary antibody was targeted at MNV-1. The reason for this finding is yet to be clarified, but it could be the specificity that exists among different MNV strains in the bacterium-binding profiles.

In terms of MNV infectivity (Figure 4), even at a low concentration, the addition of *S. maltophilia* B4 inhibited MNV infection, but *C. massiliae* F5 did not demonstrate a significant impact on MNV infectivity. Since MNV particles were not physically removed from the mixture as they were in the previous experiment, the blocking of binding sites is likely to have been responsible for the loss of MNV infectivity; once virus particles are adsorbed to bacterial cells via specific receptor–ligand interactions, the binding sites on the virus surface are occupied and, hence, no longer available for making contact with the receptor substance on the surface of the target cell and initiating the infection process, eventually leading to lowered virus infectivity. Comparing Figure 4 and Figure 5, since the addition of anti-GD1a antibody significantly restored the infectivity of MNV particles, the inhibitory effect is assumed to be attributable to GD1a ganglioside. In addition, the fact that the two anti-GD1a antibody concentration levels yielded similar results suggests that only a low level of GD1a ganglioside is present in the extracellular matrix of *S. maltophilia* B4, because its effect can be offset by a low dose of antibody, but it is highly effective in mediating the MNV–bacterium binding. However, due to the limitations of the experimental conditions in this study, further research is required to validate this hypothesis.

It is also worth noting in our study the role factors other than GD1a ganglioside play in virus–bacterium binding and suppressing the infection of MNV. In both the ELISA and plaque assay results (Figure 2 and Figure 3), MNV particles demonstrated the ability to bind to *C. massiliae* F5, the GD1a-negative strain. A similar result has been reported in a previous study in which MNV was found to be capable of binding to various commensal bacteria and fungi [62]. Although the mechanism behind such binding is yet to be fully understood, it is generally believed to be beneficial for the infection process [42,63]. Additionally, in Figure 4 and Figure 5, with or without the addition of the anti-GD1a antibody, hardly any plaque formed once the *S. maltophilia* B4 concentration reached 1.6 × 10^7^ cells/mL or higher, suggesting something other than GD1a ganglioside was either facilitating virus–cell binding or directly inhibiting MNV infection. A similar result has been reported by Taube et al. [51], who found that removing sialic acids from cultured murine macrophages by adding lectins only partly abolished the infection, indicating that additional factors may also play an important role in the virus–cell binding process. Additionally, the way in which GD1a ganglioside is present can also make a difference. Nilsson et al. [64] reported that the binding of adenovirus type 37 to corneal cells was not mediated by GD1a ganglioside itself, but rather by one or more cell surface glycoproteins that contain the GD1a glycan motif. Further investigation would help us to expand our current understanding, but progress has been impeded by factors such as the complexity of the extracellular matrix and the huge variety of substances involved.

The GD1a-positive bacterial strain identified and employed in our study, *S. maltophilia*, has been well documented for its ubiquitous occurrence in various aquatic environments [65]. In the context of wastewater, it has been isolated from granular sludge, activated sludge, and the finished water of wastewater treatment plants [66,67,68], indicating a high resistance to the treatment process. It has also been found to tolerate a wide selection of antibiotic substances, including tetracycline and heavy metals [69]. Despite it being an opportunistic human pathogen [65,70], the enrichment of *S. maltophilia* in treatment plants has been proposed for a number of potential applications, including bioremediation, biocontrol, and even medical use [71]. From the perspective of the biological water reclamation solutions, the binding ability of this bacterial strain can lead to some interesting discussions regarding a multifunctional biotreatment approach to a safer, more sustainable, and efficient water reuse scheme. The adsorption of virus particles to the biomass has been demonstrated to contribute considerably to virus removal during the wastewater treatment process [72,73,74,75]. Virus particles are generally considered difficult to remove from wastewater due to their small size, but the adsorbed or bound virus particles have a higher chance of being retained in the treatment plant by gravitational sedimentation or membrane filtration. Bacterial species capable of producing effective enteric virus receptors that also have a broad binding spectrum can be engineered to facilitate this process. The idea of a broad binding spectrum receptor has been supported by related studies. For instance, a wide range of viruses can use terminal sialic acids as attachment receptors to initiate infection [51]. In fact, in addition to the MNV that we used in the present study, GD1a ganglioside also takes part in the specific binding of adenovirus, polyomavirus, and porcine sapovirus [64,76,77]. Specific virus binding substances have also been proposed to improve virus recovery from and detection in environmental water samples, which have long been impeded by various factors such as low initial concentrations and the presence of inhibitory substances [47].

In conclusion, the presence of GD1a-positive bacteria in sewage is a promising sign for advanced water quality management due to their specific virus binding ability. The enrichment of such bacterial strains in selected locations may help to bind and capture certain species of viruses to lower their health impact. However, as pertinent research is still at an early stage and some key aspects still need to be determined, an extensive number of studies will need to be conducted in the future before the specific interactions can be put into practical application.

## 4. Materials and Methods

### 4.1. Screening Out GD1a-Positive Bacterial Strains from Sewage Sample

A serial screening approach was performed to identify and separate bacterial strains that are capable of expressing GD1a ganglioside from a sewage sample. The objective was to investigate the existence of indigenous GD1a-positive bacterial strains in a natural aquatic environment that are also culturable in vitro. In short, a selection of experiments, including flow cytometry, chemiluminescent dot blot, Western blotting, 16S RNA sequencing, and ELISA, were conducted step by step for strain selection and validation. Interested readers may refer to the Appendix A section for more details about the screening process.

### 4.2. Preparation of Media, Bacterial Suspension, and Virus Stock

The bacterial strains selected from the previous screening, one GD1a-positive and one negative, were incubated in R-2A broth for 18–24 h at 37 °C. The culture was then centrifuged (4000× *g*, 5 min) to concentrate the bacterial cells. After removing the supernatant, the pellet was resuspended with DPBS (05913; Nissui, Tokyo, Japan) and centrifuged again (4000× *g*, 5 min). The supernatant was removed, and DPBS was added to resuspend the pellet until the OD_600_ value was 1.0, which was to adjust the concentration of the bacterial suspension. The mixture was heated in a Thermomixer Comfort (Eppendorf AG, Hamburg, Germany) at 65 °C for 40 min for bacterium inactivation to avoid a potential negative effect on the RAW 264.7 cells. The inactivated bacterial suspension was used for the ELISA and plaque assays; all experiments were performed in triplicate

Two MNV strains, MNV-1 and MNV S7, were used in this study. The virus strains were added to confluent RAW 264.7 cells cultured in DMEM. DMEM was prepared using the following procedure: 4.75 g of Dulbecco’s Modified Eagle’s Medium (MEM) (05919; Nissui, Tokyo, Japan) and 1.19 g of HEPES (GB10; Doujindo, Kumamoto, Japan) were dissolved in 430 mL of MilliQ water. After autoclave disinfection, 50 mL of fetal bovine serum (26140079; Gibco, MD, USA), 7.5 mL of 7.5% sodium bicarbonate solution, 10 mL of L-glutamine (25030-081; Gibco, MD, USA), and 10 mL of penicillin–streptomycin (15140-122, Gibco, MD, USA) were added to the medium. After a 7-day cultivation, the RAW 264.7 cells were frozen and thawed three times to release virus particles. The suspension was then filtered with a 0.45 µm membrane (A045H047A; Advantec, Tokyo, Japan), and the filtrate was used as virus stock.

### 4.3. ELISA for MNV–Bacterium Binding Assessment

The wells on the ELISA plate (65506; Greiner Bio-One, Stift Kremsmünster, Netherlands) were loaded with 200 μL of inactivated bacterial suspension, and the plate was then stored at 4 °C overnight for bacterium immobilization. On the following day, the liquid was removed from each well and the wells were washed twice with 200 μL of DPBS, followed by the addition of 200 μL of 5% (*w*/*v*) BSA-PBS. The plate was stored for 2 h at room temperature, and then, the wells were washed twice with 200 μL of DPBS, followed by the addition of 100 μL of virus suspension. The plate was left for reaction for 1 h at room temperature. The wells were washed twice with 200 μL of DPBS, and 100 μL of anti-MNV antibody (MABF2097; Sigma-Aldrich, MO, USA) diluted 1:500 with 5% (*w*/*v*) BSA-PBS was added to each well as the primary antibody. The plate was stored at room temperature for 1 h before being washed twice with DPBS. The secondary antibody used in this study was goat anti-rabbit IgG H&L (HRP) (ab6721; Abcam, Cambridge, UK) diluted 1:1000 with 5% (*w*/*v*) BSA-PBS. One hundred microliters of secondary antibody was then added to each well, and the plate was stored at room temperature for reaction. After 1 h, the secondary antibody was removed, and each well was washed four times with 200 μL of DPBS. To each well was added 100 μL of an o-phenylenediamine dihydrochloride (OPD) solution made by mixing 100 mL of MilliQ water, 0.71 g of Na_2_HPO_4_, 1 tablet of OPD, 0.52 g of citric acid, and 30 μL of H_2_O_2_. The plate was left at room temperature for 30 min, and then 50 μL of 2M H_2_SO_4_ was added to each well to stop the reaction. The absorption at 490 nm, 1 s, was measured with a plate reader (ARVO; Perkin Elmer, Waltham, MA, USA). The control group was a set of parallel experiments in which the virus stock suspension was replaced by DPBS.

### 4.4. Plaque Assay in the Presence of Bacterial Components

The inactivated bacterial suspension was mixed 1:1 with MNV stock diluted to a plaque-forming concentration. The mixture was put in a shaker (Multi-Shaker oven HB; Taitec, Saitama, Japan) for 10 min, followed by centrifugation (4000× *g*, 5 min) to separate bacterial cells from the mixture. The supernatant was filtered with a 0.20 μm membrane (25AS020AS; Advantec, Tokyo, Japan) to remove residual bacterial content, and 1 mL of the permeate was added to each well on a 6-well plate with confluent RAW 264.7 cells prepared in advance. After spending 1 h in the incubator (37 °C, 5% CO_2_), the liquid was removed from the wells, and 2 mL of agar medium made of a 1:1 mixture of 2×DMEM and agar solution (01101-34; Nacalai Tesque, Kyoto, Japan) was added to each well. The mixture was left for solidification at room temperature, and then, the plate was transferred to the incubator. After a 2-day incubation, 1 mL of 0.03% neutral red solution was added to each well. After another 4 h in the incubator for staining, the neutral red solution was removed and the number of plaques was counted. The control group was a set of parallel experiments in which the inactivated bacterial suspension was added to MNV stock.

### 4.5. Viral Infectivity Assessment

To investigate whether the presence of GD1a-positive bacteria led to a change in MNV infectivity, another set of plaque assays was performed with some modifications in the setting of the test groups and sample treatments. In this case, the mixture of inactivated bacterial suspension and MNV stock was not centrifuged. Instead, it was directly added to the RAW 264.7 cell culture. A four-fold serial dilution of inactivated bacterial suspension was introduced to assess the effects of different bacterial concentrations. Finally, to investigate if extracellular substances other than GD1a ganglioside also contributed to the inhibitory effect on MNV infectivity, a set of plaque assays with the addition of anti-GD1a antibody was performed using *S. maltophilia* B4 and MNV S7. The experimental protocol was largely the same as the one introduced in Section 4.4, but prior to inoculation, the inactivated bacterial suspension was added with anti-GD1a antibody (MAB5606; EMD Millipore, Billerica, MA, USA) at two concentration levels (0.02 and 2.5 μg/mL).

### 4.6. Statistical Analysis

Student’s *t*-tests were performed with Microsoft Excel for Windows 2016 (Microsoft Corp, Redmond, WA, USA) to determine whether the relative adsorption values and the virus infectious titer values in the different groups were significantly different. A *p* value of 0.05 or lower was considered significant.

## Figures and Tables

**Figure 1 molecules-25-04084-f001:**
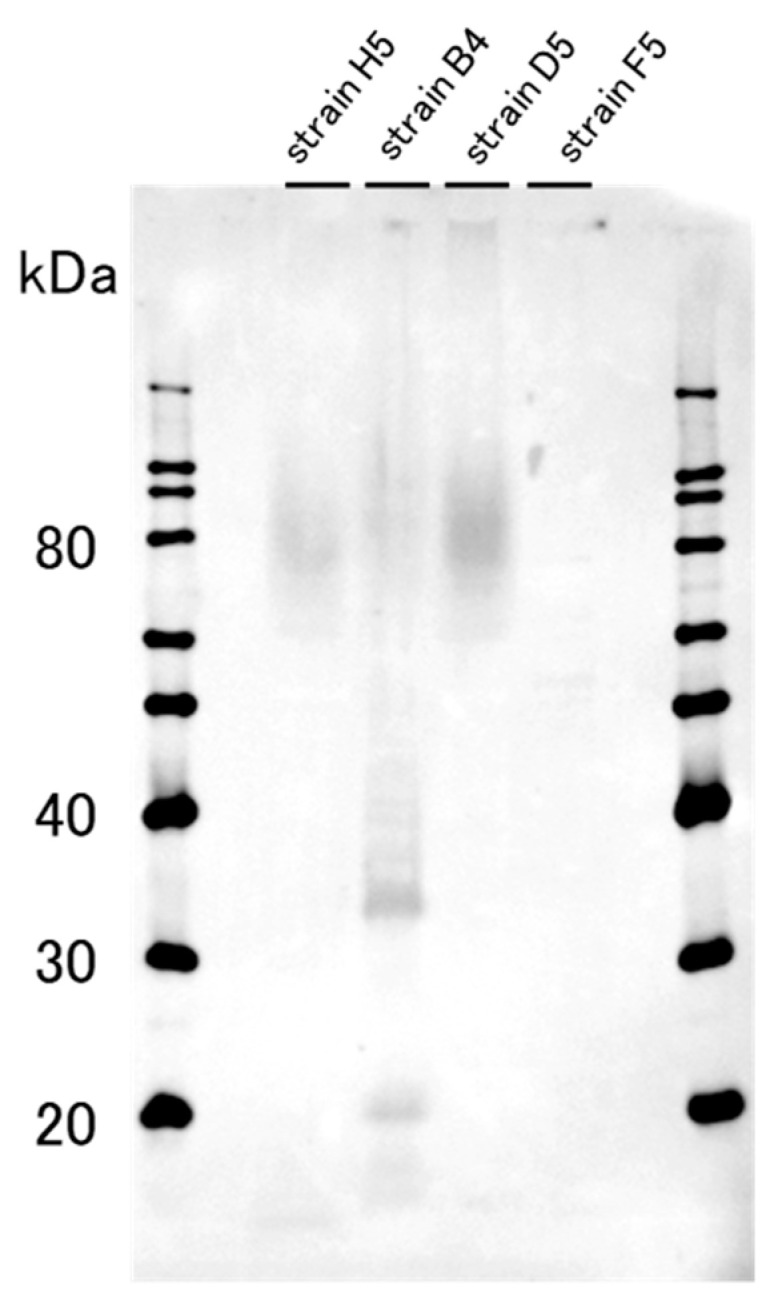
Western blotting results for four bacterial strains. GD1a ganglioside was detected with the anti-GD1a ganglioside antibody. Strain B4 showed two relatively clear bands at around 20 and 33 kDa and was considered as GD1a-positive. Strains H5 and D5, on the other hand, were not used in the subsequent experiments, as only smeared bands were observed.

**Figure 2 molecules-25-04084-f002:**
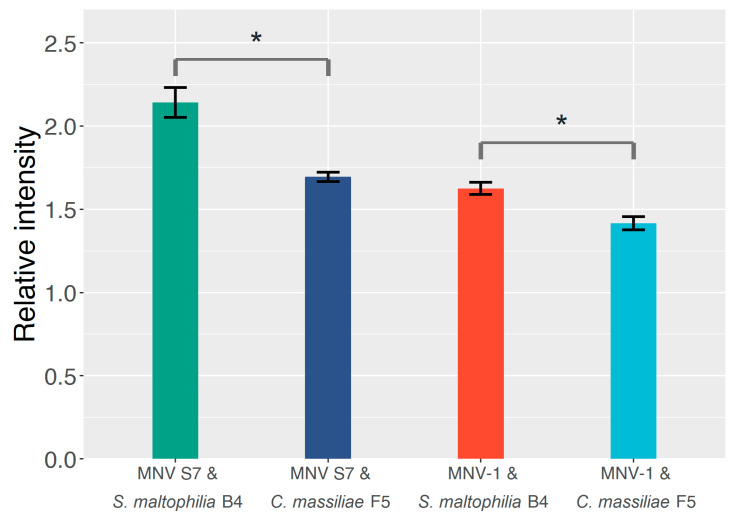
The ELISA results for the four bacterium–murine norovirus (MNV) combinations. The extent of binding is measured according to the relative adsorption intensity, calculated by dividing the absorption values of the test group by those of the control group. The heights of the main bars and error bars represent the average values and standard deviations from three independent parallel experiments. *: *p* < 0.05.

**Figure 3 molecules-25-04084-f003:**
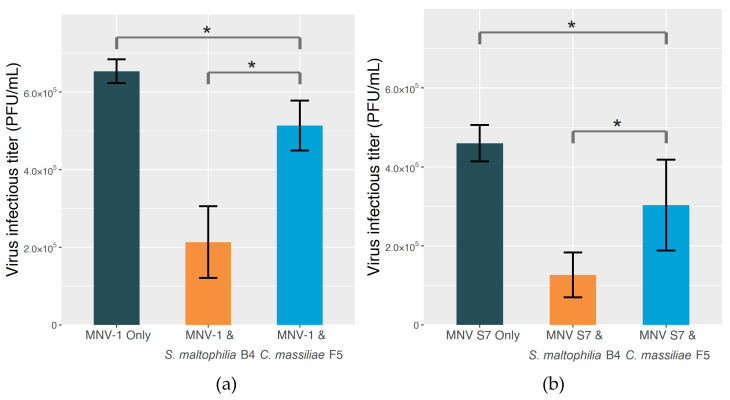
The plaque assay results for MNV with/without the addition of a bacterial strain. Prior to inoculation, centrifugation was performed as a measure to remove the virus particles bound to bacterial cells from the mixture. The extent of virus–bacterium binding is therefore reflected by the drop in plaque-forming unit (PFU) count. (**a**) Infectious titers of MNV-1 without either of the strains, MNV-1 with *S. maltophilia* B4, and MNV-1 with *C. massiliae* F5. (**b**) Infectious titers of MNV S7 without either of the strains, MNV S7 with *S. maltophilia* B4, and MNV S7 with *C. massiliae* F5.

**Figure 4 molecules-25-04084-f004:**
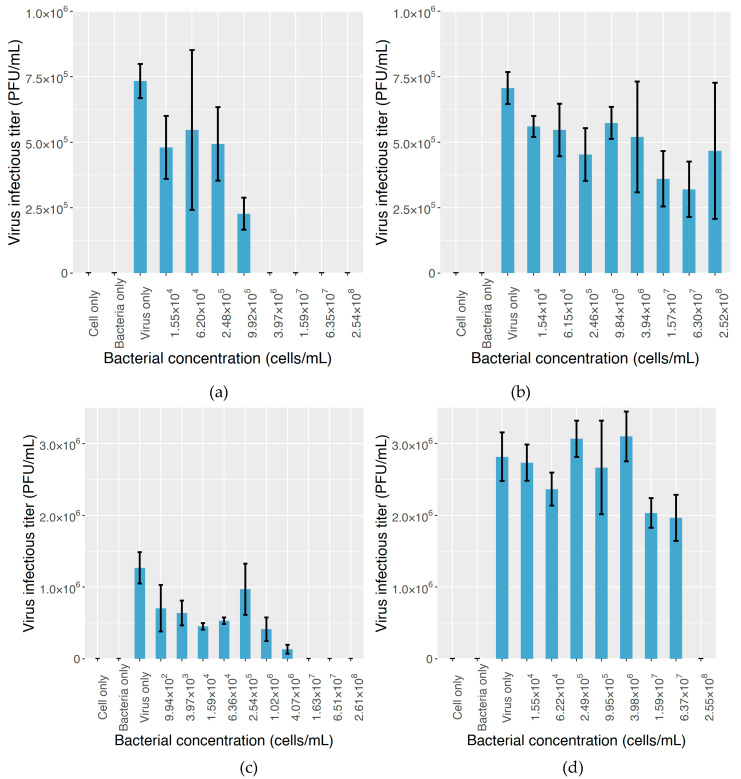
The plaque assay results for the inhibitory effect of bacterial cells on MNV infectivity. The mixture of bacteria and virus suspension was directly added to RAW cell culture without centrifugation. The extent of the inhibitory effect is demonstrated by the drop in infectious titer as the bacterial content increases. (**a**) MNV-1 mixed with *S. maltophilia* B4. (**b**) MNV-1 mixed with *C. massiliae* F5. (**c**) MNV S7 mixed with *S. maltophilia* B4. (**d**) MNV S7 mixed with *C. massiliae* F5. In the experiment, three control groups were introduced: “cell only”: neither the virus stock nor bacterial suspension was added to the cell culture; “bacteria only”: only the bacterial suspension was added to the cell culture; “virus only”: only the virus stock was added to the cell culture.

**Figure 5 molecules-25-04084-f005:**
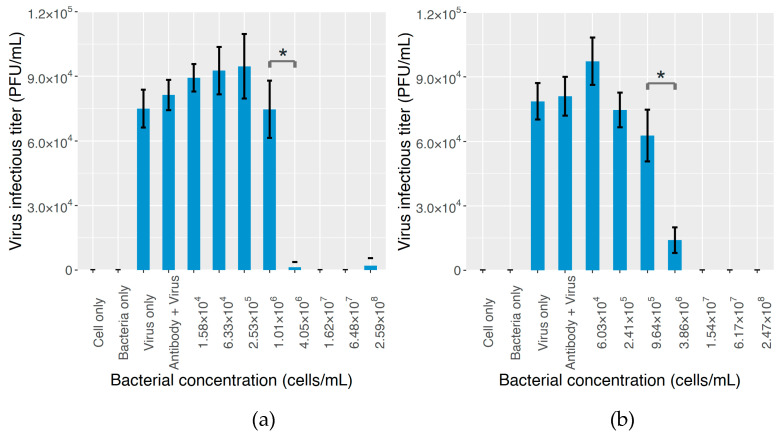
The plaque assay result for the inhibitory effect of bacterial cells on MNV infectivity with the addition of the anti-GD1a antibody. MNV S7 and *S. maltophilia* B4 were used as the virus and bacterial strain, respectively. The addition of the anti-GD1a antibody was aimed at abating the efficacy of GD1a ganglioside expressed by bacteria, and the disturbed virus–bacterium binding was reflected in the lack of impact on MNV infectivity from bacterial content. (**a**) 0.02 µg/mL antibody was added to the bacterial suspension. (**b**) 2.5 µg/mL antibody was added to the bacterial suspension. Four control groups, “cell only”, “bacteria only”, “virus only”, and “antibody + virus”, were introduced. The meanings of the first three are explained in Figure 3, and in the group of “antibody + virus”, only the anti-GD1a antibody and the virus stock were added to the cell culture.

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
