# Peer review of "The Effect of GD1a Ganglioside-Expressing Bacterial Strains on Murine Norovirus Infectivity"

_molecules, 2020, doi:10.3390/molecules25184084_

Round 1
Reviewer 1 Report
Zhu et al tried to make links between bacterial GD1a ganglioside and norovirus infectivity. They found GD1a positive bacteria bind better to MNV and this binding compromises MNV infectivity. The significance of this finding is not clearly stated. Is more GD1a positive bacteria were encouraged in sewage? The manuscript can be improved by concise and clear presentation. Statistics should be applied and labelled in figures with number “n” indicated.
Author Response
Response to Reviewer 1 Comments
We sincerely thank reviewer 1 for the constructive comments. The points mentioned have been carefully taken into consideration, please see our response below.
Point 1: The significance of this finding is not clearly stated. Is more GD1a positive bacteria were encouraged in sewage? The manuscript can be improved by concise and clear presentation.
Response 1: agree. Our conclusion is that although GD1a-positive bacteria have the potential to be a virus binding agent in the aquatic environment, the applicability still needs further validation, and our study should serve as a preliminary step. The manuscript has been revised to better elaborate on our opinion and offer clarity. Please see page 9, line 317-321.
Point 2: Statistics should be applied and labelled in figures with number “n” indicated.
Response 1: agree. We have revised Figures 2, 3a, 3b, 5a, and 5b to add significance level, related expression has also been modified. As for number "n", we have included the information about the experimental replication in the figure description (page 4, line 152) and Materials and Methods section (page 10, line 342), therefore it is not repeated in the figures.
Reviewer 2 Report
The research: The effect of GD1a ganglioside-expressing bacterial strains on murine norovirus infectivity by Yifan Zhu et al. is very interesting and gives new information.
Comments on the paper:
Title reflects the paper’s content.
Abstract is appropriate.
Introduction
The introduction makes a proper introduction to the subject matter of the paper.
The study was carried out using well chosen methods with guarantee the reliability of the results obtained.
Results and Discussion
Well written.
Editoral comments:
Page 2, line 54 should be Pang et al. [21]
Page 2, line 59 should be Eftim et al. [22]
Page 3, line 99 should be Li et al. [53]
Page 8, line 237 should be Amarasiri et al. [60]
Page 9, line 289 should be Nilsson et al. [65]
Page 9, line 331 should be (4000×g, 5 min)
Page 9, line 333 should be (4000×g, 5 min)
Author Response
Response to Reviewer 2 Comments
We thank the reviewer 2 for the positive comments.
All the format-related issues have been addressed, please check the following lines in the revised manuscript.
Page 2, line 54
Page 2, line 59
Page 3, line 99
Page 8, line 238
Page 9, line 290
Page 9, line 332
Page 9, line 334
Round 2
Reviewer 1 Report
The manuscript has been improved. Moderate language editing is recommended.